# Positional Biases Shift as Inputs Approach Context Window Limits

**Blerta Veseli[1], Julian Chibane[2], Mariya Toneva[3], Alexander Koller[1]**
[1] Saarland Informatics Campus, Saarland University, Germany
[2] Max Planck Institute for Informatics, Saarland Informatics Campus, Germany
[3] Max Planck Institute for Software Systems, Saarland Informatics Campus, Germany

## Abstract

Large Language Models (LLMs) often struggle to use information across long inputs effectively. Prior work has identified positional biases, such as the Lost in the Middle (LiM) effect, where models perform better when information appears at the beginning (primacy bias) or end (recency bias) of the input, rather than in the middle. However, long-context studies have not consistently replicated these effects, raising questions about their intensity and the conditions under which they manifest. To address this, we conducted a comprehensive analysis using relative rather than absolute input lengths, defined with respect to each model's context window. Our findings reveal that the LiM effect is strongest when inputs occupy up to 50% of a model's context window. Beyond that, the primacy bias weakens, while recency bias remains relatively stable. This effectively eliminates the LiM effect; instead, we observe a distance-based bias, where model performance is better when relevant information is closer to the end of the input. Furthermore, our results suggest that successful retrieval is a prerequisite for reasoning in LLMs, and that the observed positional biases in reasoning are largely inherited from retrieval. These insights have implications for long-context tasks, the design of future LLM benchmarks, and evaluation methodologies for LLMs handling extended inputs.

## 1 Introduction

Large language models (LLMs) are increasingly leveraged in real-life scenarios, facing tasks such as search (Kelly et al., 2023; Ziems et al., 2023), coding (Nam et al., 2024; Sun et al., 2025) or summarization (Chang et al., 2024; Zhang et al., 2024a). Beyond their in-weight knowledge, they are typically provided task-specific information at inference time. To successfully perform such tasks, LLMs require retrieval and reasoning capabilities to a) find relevant information and b) infer over the retrieved information to satisfy the user's queries. Since relevant information for the user query can span large web search results, legal documents, book excerpts, and the like, this calls for LLMs that are capable of successfully retrieving and reasoning over excessively large inputs. While in recent years, the context windows of LLMs have grown substantially, their ability to fully exploit these extended windows does not scale proportionally.

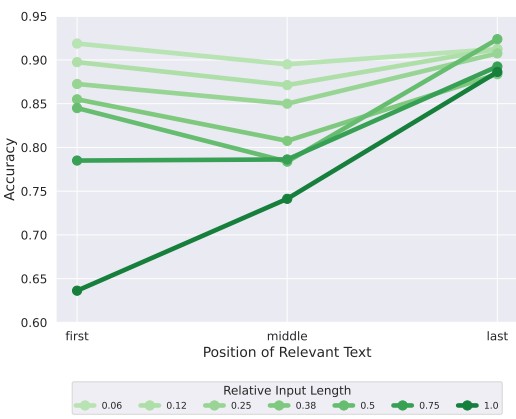

Figure 1: The "Lost in the Middle" (LiM) effect describes how models favor information from the beginning (primacy bias) and end (recency bias) over the middle of an input. Our findings reveal that as inputs reach a model's context window size, the primacy bias drops and the LiM effect disappears. We show this effect across models.

Understanding how LLMs make use of long inputs has been an active question in NLP. Notably, Liu et al. (2024) shed new light on this by identifying positional biases — specifically, primacy and recency biases — where models have improved accuracy of retrieving information positioned at the beginning and end of an input. This leads to the Lost in the Middle (LiM) effect, in which model performance declines when relevant information is located in the middle of an input. This effect, however, has not been consistently observed – research using different LLMs has not found these LiM effects (Zhang et al., 2024b). This naturally leads to the question: *What conditions underly the Lost in the Middle effect: where, when, and for what type of models and data does it occur?*

When examining this question and the previous literature, we observed that prior works not only differ in the models that they evaluated, but also differ drastically in the length of inputs that were leveraged for evaluation. The study by Liu et al. (2024), along with others (Hsieh et al., 2024; Liu et al., 2024), that demonstrated the LiM effect, evaluated sequences of up to 6K tokens, while Zhang et al. (2024b), using approximately 100K-token inputs, could not confirm the LiM effect. This observation hints at one aspect of the LiM-favorable conditions, leading to a more precise question: *How does an LLM's input length interact with its positional biases?*

We address these research questions by introducing a relative input length, $L_{rel}$, which represents the ratio of an input's length to the model's context window size. Previous positional analyses did not completely max out a model's context window but focused on shorter and less expensive [1] inputs (e.g., less than 50% of the context window for 8K models and under 10% for models with over 32K tokens). Instead, we investigate position effects across the full context window. We place relevant information at different positions in the input text (first, middle, last) and pad it with irrelevant information following Levy et al. (2024). Additionally, we define simple yet concrete metrics to measure primacy, recency, and lost-in-the-middle biases directly from the data.

In addition to retrieval tasks, we investigate reasoning tasks and observe similar positional biases. To examine whether retrieval is a necessary precursor to reasoning in LLMs, we analyze reasoning success probabilities conditioned on successful versus unsuccessful retrieval. Our findings confirm that successful retrieval is indeed a prerequisite for effective reasoning, and when retrieval fails, reasoning performance drops significantly. Moreover, we find that positional biases in reasoning largely stem from those in retrieval; conditioning on successful retrieval largely eliminates these biases.

Our approach enables four key findings:

- First, in contrast to inconsistent findings in the literature, we find the LiM effect is prominent across all tested models with different context window sizes.

- Second, by evaluating LiM with respect to a model's maximum context window size, we find that LiM is present for all models when $L_{rel} \leq 0.5$. These findings reconcile inconsistent earlier findings: works that tested LiM on shorter inputs observed it because the inputs made up less than 50% of the context widow, whereas works that focused on long-context benchmarks exceeding 100K tokens did not detect LiM, as the effect tends to vanish or diminish significantly for inputs that occupy a larger portion of the context window.

- Third, we characterize why the LiM effect vanishes beyond 50% of an LLM's context size. A simple scenario could be that overall performance gets closer to chance levels across all positions as input length increases, as found in Liu et al. (2024), which can mask position-specific differences. However, our findings indicate that the disappearance of the LiM effect is largely driven by a strong decrease in favoring initial positions as input length grows (i.e., a decrease in primacy bias) – we observe that accuracy for initial positions drops to the same level as, or even below, that of middle positions.

---

[1] For example, running our full set of experiments on GPT-4 Turbo (128K) cost approximately 10,000 US dollars, while evaluating an open-source model such as Llama 3.1 70B (128K) requires roughly 24 hours on four H100 GPUs (each with 80GB VRAM) for just one dataset.

- We find that successful retrieval is a crutial precursor to effective reasoning in LLMs; when retrieval fails, reasoning performance drops significantly. Moreover, we show that positional biases in reasoning are largely inherited from those in retrieval.

## 2 Related Work

**Long-Context Benchmarks use Absolute Input Lengths.** While LLMs are increasingly equipped with larger context windows, their ability to efficiently process long inputs does not scale proportionally. Recent benchmarks – ranging from realistic datasets (Karpinska et al., 2024; An et al., 2024; Zhang et al., 2024b; Bai et al., 2024; Kočiský et al., 2018; Kryscinski et al., 2022; Shaham et al., 2023) to synthetic ones (Hsieh et al., 2024; Song et al., 2025; Yuan et al., 2024; Modarressi et al., 2025; Tian et al., 2025; Wang et al., 2024; Kuratov et al., 2024) – aim to evaluate long-context capabilities. While most of these benchmarks contain only sequences of predefined lengths, others (Hsieh et al., 2024; Song et al., 2025; Wang et al., 2024) allow adapting input length as needed. Existing benchmarks often evaluate models with different context window sizes using the same absolute input lengths. This creates an imbalance – for example, a 32K-token input may fully saturate one model's context but occupy only a fraction of another's – making it hard to compare how well models use their available capacity. To address this, we define input length relative to context window size, enabling more meaningful comparisons and insights into scaling behavior.

**Inconsistent Insights about Positional Biases.** Models often exhibit positional biases, such as performing better on information at the beginning (primacy) or end (recency) over the middle (Lost in the Middle, LiM) (Ivgi et al., 2023; Liu et al., 2024; Kamradt, 2023). But findings on positional biases are inconsistent (Bai et al., 2024; Modarressi et al., 2025; Hsieh et al., 2024; Wang et al., 2024; Tian et al., 2025; Song et al., 2025; Zhang et al., 2024b). Some studies report no LiM effect (Zhang et al., 2024b; Song et al., 2025), while others observe it only at a single input length among many tested (Modarressi et al., 2025). Primacy bias has been consistently observed in tasks where models are presented with a list of candidate answers (Wang et al., 2024; 2023), with Wang et al. (2024) further noting that the effect intensifies with input length for some models. However, these studies differ substantially in input lengths and tasks tested. For example, Modarressi et al. (2025) evaluate tasks that require substantial reliance on in-weight knowledge; Song et al. (2025) use multi-evidence setups that distribute information evenly across the input; and only Zhang et al. (2024b) tests input lengths beyond 100K tokens. By contrast, studies on shorter contexts ($\leq 6K$ tokens) and more controlled synthetic tasks consistently find positional biases (Levy et al., 2024; Liu et al., 2024; Xu et al., 2024). To address this, we normalize input lengths and employ controlled tasks that (1) present relevant information in a single location at a time, and (2) minimize reliance on in-weight knowledge, isolating positional biases under input usage.

**Retrieval Effects Reflected in Reasoning Results.** To improve long-context reasoning, several studies have proposed *retrieve-then-think* approaches that show substantial gains by encouraging models to first output the relevant information and then the inferred answer. Some approaches rely on prompting strategies (Cattan et al., 2024; Li et al., 2024b;a), similar to CoT (Wei et al., 2022) or analogical prompting (Yasunaga et al., 2024), while others improve performance by for example fine-tuning (Qiu et al., 2025). At the same time, these works find that when models did not output the correct facts first, this results in reasoning errors (Li et al., 2024a; Levy et al., 2024). Although these results indicate that retrieval is important for reasoning, improvements might stem from decomposing a complex single-step task into simpler substeps. Moreover, prior works so far find positional biases for retrieval and reasoning separately and do not show their connection. To address this, we construct retrieval-reasoning minimal pairs, where each retrieval question targets the exact information needed for the corresponding reasoning question. This enables assessing reasoning performance under retrieval success or failure. We find that positional biases in reasoning are largely inherited from those in retrieval.

| Dataset | Premises | Reasoning Question | Retrieval Question | GT |
|---|---|---|---|---|
| MonoRel | Julie is younger than Julian. Samantha is younger than Julie. | Is Samantha younger than Julian? | Is Julie younger than Julian? | ✓ |
| | Jimmy is slower than Michael. Scott is slower than Jimmy. | Is Michael slower than Scott? | Is Michael slower than Jimmy? | ✗ |
| PIR | John's kitchen is marble-floored. Ethan is in John's kitchen. | Is Ethan in a marble-floored room? | Is John's kitchen marble-floored? | ✓ |
| | Michael is in Anna's great hall. Anna's great hall is green-walled. | Is Michael in a white-walled room? | Is Anna's great hall white-walled? | ✗ |
| RuleTaker | If X is furry and X is good then X is tall. Erin is furry. Erin is good. | Erin is tall. | Erin is furry and good. | ✓ |
| | If X is tall and X is good then X is flat. Fiona is lazy. Fiona is tall. | Fiona is flat. | Fiona is tall and good. | ✗ |
| BoxTracker | Ella found two boxes: Box 0 contains the wire. Box 1 contains the ball. Ella puts the key into Box 1. Ella moves the contents from Box 1 to Box 0. | After performing all operations, Box 1 contains nothing. | The key is put into Box 1. | ✓ |
| | Hannah found two boxes: Box 0 contains the wire. Box 1 contains the ball. Hannah moves the contents from Box 0 to Box 1. Hanna puts the hat into Box 0. | After performing all operations, Box 1 contains the wire and the ball and the hat. | The wire is put into Box 1. | ✗ |

Table 1: **Datasets Overview.** Each row illustrates a representative instance from a dataset, consisting of a set of premises, a corresponding pair of retrieval and reasoning questions, and a ground truth (GT) label – where a checkmark (✓) denotes a true-labeled example and a cross (✗) denotes a false-labeled one. Each retrieval question targets the specific information needed to answer its paired reasoning question. While the retrieval question is directly answered by one of the premises, the reasoning question requires combining multiple premises to infer an unstated conclusion. All examples are shortened for brevity. See full examples in Appendix A.1.

## 3   Putting Positional Biases in Relation to Context Window Size

LLMs exhibit positional biases, such as performing better on information at the beginning (primacy) or end (recency) over the middle (Lost in the Middle, LiM) (Ivgi et al., 2023; Liu et al., 2024; Kamradt, 2023). Prior work is *a)* inconsistent about if and when positional biases occur and *b)* fails to connect positional biases in retrieval with those in reasoning. We address *b)* by constructing a dataset of Retrieval-Reasoning Minimal Pairs (Section 3.1), where models can not rely on in-weight knowledge but have to use the text input, and where the same atomic (sentence) information is needed for the retrieval and reasoning task. To adress *a)* we propose to measure model accuracy with respect to *relative* input length, i.e., the length of the input normalised by the model's context window size (Sec. 3.2). We propose concrete metrics to measure primacy, recency, and Lost in the Middle (LiM) biases in Sec. 3.3, and give an overview of evaluated models in Sec. 3.4.

### 3.1   Datasets: Retrieval-Reasoning Minimal Pairs

In our analysis, we use four reasoning datasets: MonoRel, PIR, (Simplified) RuleTaker by Levy et al. (2024), and an entity tracking dataset (BoxTracker) by Kim & Schuster (2023) (see Table 1). MonoRel and PIR each consist of two premises. MonoRel consists of comparative statements expressing relationships among three people (A, B, and C). For example: "A is slower than B, B is slower than C" with the corresponding reasoning question being: "Is A slower than C?". The comparative adjective varies (e.g., slower, younger, taller, faster,

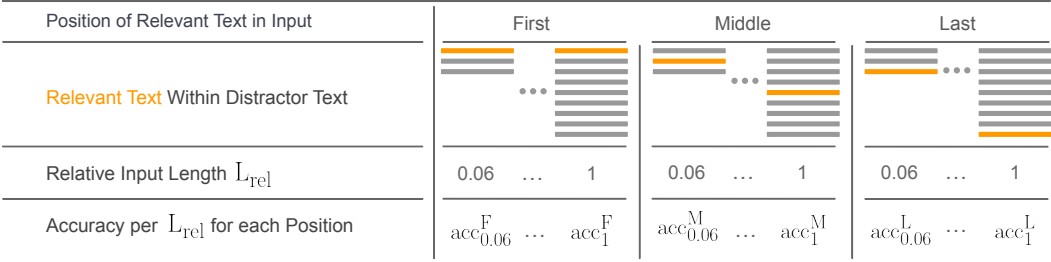

| Position of Relevant Text in Input | First | | Middle | | Last | |
|---|---|---|---|---|---|---|
| Relevant Text Within Distractor Text | | | | | | |
| Relative Input Length $L_{\text{rel}}$ | 0.06 ... 1 | | 0.06 ... 1 | | 0.06 ... 1 | |
| Accuracy per $L_{\text{rel}}$ for each Position | $\text{acc}^{\text{F}}_{0.06}$ ... $\text{acc}^{\text{F}}_{1}$ | | $\text{acc}^{\text{M}}_{0.06}$ ... $\text{acc}^{\text{M}}_{1}$ | | $\text{acc}^{\text{L}}_{0.06}$ ... $\text{acc}^{\text{L}}_{1}$ | |

Figure 2: **Method Overview.** Schematic overview of how relevant text (orange) is positioned within distractor text (gray) at three locations (first, middle, last) across varying input lengths (6% to 100% of the context window). For each position $P \in \{$First, Middle, Last$\}$ and each relative input length $L_{rel} \in \{0.6, 0.12, 0.25, 0.38, 0.5, 0.75, 1\}$, we measure accuracy $\text{acc}^{P}_{L_{rel}}$, illustrating how well the model performs when relevant information is placed at position $P$ for a relative input length $L_{rel}$.

etc.). PIR uses spatial premises involving a location $L$, a person $P$, and an attribute $A$, such as: "Location $L$ possesses a certain attribute $A$; person $P$ is at that location $L$." It then poses questions like: "Is person $P$ at a location with attribute $A$?". RuleTaker includes two factual premises and one logical rule, requiring the model to apply the rule to infer a conclusion based on the two facts. BoxTracker consists of sequences of four statements: two that describe the initial world state (e.g., the contents of each box), and two that describe actions altering that state (e.g., moving or inserting objects). The goal is to track how the contents of a given box change over time.

In addition, we constructed a retrieval question in correspondence to each reasoning question, creating reasoning-retrieval minimal pairs. Each retrieval question targets a specific piece of information required for the reasoning process. The answer to the retrieval question is stated explicitly in one of the premises and requires no inference. In summary, we end up with four datasets, each containing 100 instances, with two questions (retrieval and reasoning) per instance.

Following Levy et al. (2024), premises are embedded into short narratives to produce more cohesive and natural-looking paragraphs. The result is referred to as relevant text in the following. MonoRel, PIR, and RuleTaker are already formulated as balanced binary classification tasks for reasoning (50% true-labeled, 50% false-labeled), and we extend this balance to the corresponding retrieval questions in our minimal retrieval-reasoning pairs. We also adapt the BoxTracker dataset to follow the same format for consistency (see Table 1).

### 3.2 Method: Varying Input Length and Positions

To investigate positional biases, we systematically vary two factors: the input length and the position of relevant text within that input.

We define relative input length $L_{\text{rel}}$ as a proportion of the input text length $L_{\text{input}}$ to a model's context window size $L_{\text{max}}$:

$$L_{\text{rel}} = \frac{L_{\text{input}}}{L_{\text{max}}}. \tag{1}$$

For our experiments we choose $L_{\text{rel}} \in \{0.06, 0.12, 0.25, 0.38, 0.5, 0.75, 1\}$, aligned with prior work Levy et al. (2024), whose absolute input lengths $(500, 1000, 2000, 3000$ tokens$)$ map to relative values (0.06, 0.12, 0.25, 0.38) given a 8K-token context window. This allows for more direct comparison with related work. We also tested evenly spaced values (e.g., $0.1, 0.2, 0.4, 0.6, 0.8, 1$); see Figure 11, Appendix A.2.

The relevant text (see section 3.1) is then extended to the target length $L_{\text{input}} = L_{\text{rel}} \times L_{\text{max}}$ by padding with distractor text. We adopt the padding variants proposed by Levy et al.

| Model | Context Window | MonoRel | | PIR | | RuleTaker | | BoxTracker | |
|---|---|---|---|---|---|---|---|---|---|
| | | RT | RA | RT | RA | RT | RA | RT | RA |
| Llama-3.1-70B | 128K | 1.0 | 1.0 | 1.0 | 1.0 | 0.98 | 0.65 | 1.0 | 0.60 |
| Llama-3.3-70B | 128K | 1.0 | 1.0 | 1.0 | 1.0 | 0.98 | 0.73 | 1.0 | 0.88 |
| Llama-3-70B | 8K | 1.0 | 0.98 | 1.0 | 1.0 | 0.77 | 0.63 | 0.88 | 0.86 |
| Mistral-Small-24B | 32K | 1.0 | 1.0 | 1.0 | 1.0 | 1.0 | 0.91 | 0.96 | 0.78 |
| Qwen-2.5-32B | 32K | 0.95 | 0.92 | 1.0 | 0.98 | 0.97 | 0.89 | 0.97 | 0.92 |
| Gemma-2-27B | 8K | 1.0 | 1.0 | 1.0 | 1.0 | 0.96 | 0.88 | 0.87 | 0.83 |

Table 2: **Models Overview.** Here we show an overview over the models and their accuracy on base instances for reasoning (RA) and retrieval (RT). Models are selected based on their performance on base instances and with the goal to test as many different context window sizes as possible.

(2024) but omit the duplicate padding variant, which extends inputs by repeating the same instance multiple times. Since this places the same relevant information at multiple positions simultaneously, it would undermine our position-sensitive analysis. To then analyze positional effects, we place the relevant text at the beginning $F$, middle $M$, or end $L$ of the input sequence. Figure 2 shows an overview of the full approach.

### 3.3 Metrics: Quantifying Positional Biases

To measure the LiM effect as well as primacy and recency biases, we define the following metrics by making use of the definition of each positional bias. The LiM effect is present when accuracy for middle positions $acc^M$ is lower than accuracies for first $acc^F$ and last positions and $acc^L$. Accordingly, we define the metric *LiMi* (LiM intensity) as follows:

$$ LiMi = \begin{cases} \left(acc^F - acc^M\right) + \left(acc^L - acc^M\right), & \text{if } acc^F > acc^M \text{ and } acc^L > acc^M, \\ 0, & \text{otherwise.} \end{cases} \quad (2) $$

LiMi is designed such that it equals zero if the LiM effect is not present. Similarly, we define PriMi (primacy bias intensity) and ReCi (recency bias intensity) as follows:

$$ PriMi = acc^F - acc^M \quad (3) \qquad\qquad ReCi = acc^L - acc^M \quad (4) $$

Unlike LiMi, PriMi, and ReCi can have negative values. Negative values for these biases indicate a decrease in the bias.

To compute LiMi, PriMi, and ReCi relative to context window size, we use the position-specific accuracy values $acc^F$, $acc^M$, and $acc^L$ per relative input length $L_{rel}$ (see Figure 2, last row) to compute one LiMi, one PriMi and one ReCi value per $L_{rel}$.

### 3.4 Models

We select a diverse set of open-source LLMs to evaluate positional bias (see Table 2). Our primary selection criteria are: (1) coverage of a wide range of context window sizes (from 8K to 128K), and (2) sufficient performance on non-padded instances (i.e., accuracy over baseline $> 0.5$). The first criterion ensures that findings generalize across varying context window sizes, while the second ensures meaningful signal during the positional bias evaluation. We use only the instruction-tuned versions of these models. We use only instruction-tuned versions of these models.

Our model selection supports comparisons not only across varying context window sizes, but also along two axes: (1) model generation, e.g., Llama-3.1-70B vs. Llama-3.3-70B (both 128K context, same size), (2) model family, e.g., Mistral-Small-3-24B vs. Qwen2.5-32B (both 32K context, similar size). Table 2 summarizes the tested models, their context windows, and

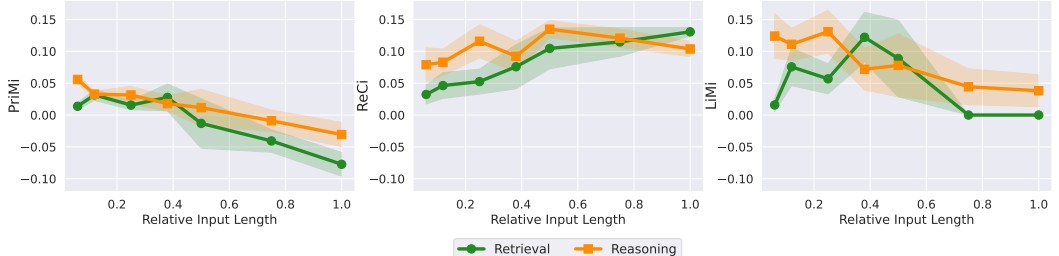

Figure 3: **Quantitative positional bias evaluation.** Visualization of positional bias intensities across varying input lengths $L_{\text{rel}}$, expressed as a proportion of the model's context window. The primacy bias intensity (PriMi) is shown in the left subplot, recency bias intensity (ReCi) in the middle, and LiM intensity (LiMi) on the right. Retrieval results are shown in green, and reasoning results in orange. Solid lines indicate the mean positional bias intensity across datasets and models, with shaded areas denoting the standard error of the mean (SEM). We observe that primacy bias decreases, while recency bias increases, as $L_{\text{rel}}$ grows. Furthermore, the LiM effect is most pronounced for $L_{\text{rel}} \leq 0.5$, peaking around $L_{\text{rel}} \approx 0.25$ for reasoning and $L_{\text{rel}} \approx 0.38$ for retrieval.

performance on non-padded inputs. To assess the effect of model size, we include results for Llama-3.1-8B (128K context) in Appendix A.4, Figure 12, enabling direct comparison to Llama-3.1-70B while keeping the main paper focused on higher-capacity models.

## 4 Findings

In this section, we analyze how positional biases evolve as input length approaches the model's context window limit. Up to 50% of the context window, we observe a more pronounced Lost in the Middle (LiM) effect, while the primacy bias decreases and the recency bias increases as input length approaches context window size (Section 4.1). Beyond 50% of a model's context window, primacy bias fades significantly, introducing a distance-based bias favoring information closer to the end of an input. An effect that makes LiM disappear (Section 4.2). Finally, our results indicate that reasoning relies on successful retrieval and that positional biases in reasoning are substantially inherited from retrieval (Section 4.3).

### 4.1 Positional Biases Emerge Consistently Relative to Context Window Size

In this section, we analyze whether positional bias trends remain consistent across models when input length is normalized by each model's context window (relative input length).

**Evaluation.** Following Sec. 3, we evaluate the six selected models on our datasets. We observe similar trends across datasets and models, so we present averaged results in Figure 3. We analyze retrieval and reasoning separately. Figure 3 shows the resulting PriMi, ReCi, and LiMi curves for retrieval and reasoning.

**Results.** From Figure 3, the LiM effect (LiMi) is strongest when $L_{\text{rel}} \leq 0.5$, across models with varying context window sizes. Beyond this relative input length, the LiM effect declines sharply. This finding helps reconcile earlier research: works that tested LiM on shorter sequences possibly observed it because they stayed below the 50% threshold, whereas long-context benchmarks exceeding 100K tokens did not detect it, as the effect tends to vanish or diminish significantly for inputs that occupy a larger portion of the context window. The primacy bias (PriMi) remains relatively stable up to $L_{\text{rel}} \approx 0.5$, after which it begins to drop steeply. In the case of retrieval, it even falls below zero at $L_{\text{rel}} = 0.5$, indicating that accuracy for first-positioned information becomes lower than for middle-positioned information. The recency bias (ReCi) exhibits a mirrored pattern: its intensity increases steadily up to $L_{\text{rel}} = 0.5$, then levels off beyond that point. Given LiMi's definition (Eq. 2), its decline with

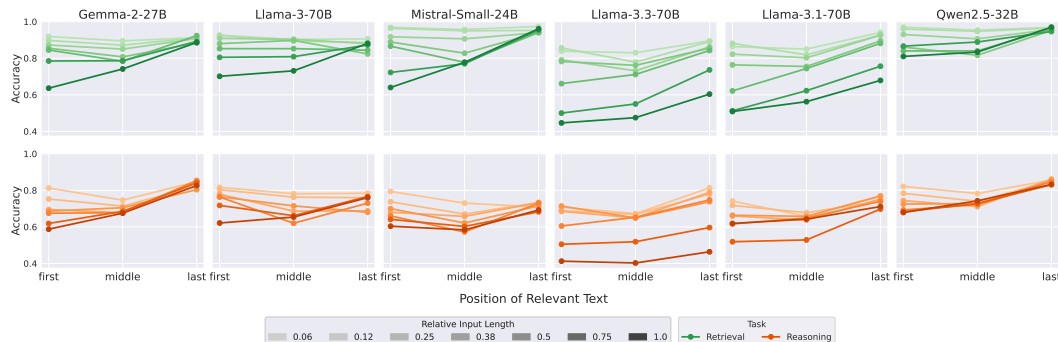

Figure 4: **Reason for disappearing LiM.** Different models are depicted in columns and tasks in rows, the x-axis represents the relevant information position, y-axis represents task accuracy. Again, we find an accuracy minima in the middle (LiM effect) of shorter inputs (light red curves) $L_{\text{rel}} \leq 0.5$. We see the LiM effect disappearing when relative input length $L_{\text{rel}}$ gets longer (darker res curves) $L_{\text{rel}} \geq 0.5$. For long sequences (dark red), task accuracy at the end stays consistent (consistent recency bias), while accuracy at the beginning drops significantly (disappearing primacy bias), removing the LiM and concluding in a distance-based bias. We show the results per dataset in Appendix A.3.

input length could stem from edge accuracies dropping to match the middle, or both falling below it, a "found in the middle" effect. Prior work favors the first hypothesis: overall accuracy drops with longer inputs (Levy et al., 2024). Yet, trends in primacy and recency suggest a third possibility: only the first position degrades, while the last holds. Since PriMi drops below zero, the fading LiM effect likely actually reflects a loss of primacy bias.

In the next section, we analyze why the LiM effect fades as input length approaches the context window limit and how this interacts with other positional biases.

## 4.2 Positional Biases Shift Toward Distance-Based Bias with Increasing Input Length

In Section 4.1, we observed that LiM effect diminishes beyond 50% of a model's context window, while the primacy bias decreases and the recency bias increases as relative input length grows. Here, we show what happens to positional biases after the threshold of 50% and explore why LiM diminishes–offering insights into how models utilize their inputs as they approach the limits of their context windows.

**Evaluation.** We present the results per model, averaged across the datasets in Fig. 4,(instead of averaging additionally across models, as in Fig. 3), and perform a more fine-grained result analysis next.

**Results.** Our findings indicate that the diminishing LiM effect after $L_{\text{rel}} \approx 0.5$ is largely driven by a weakening primacy bias – accuracy at the initial position drops to match or fall below that of the middle. As shown in Figure 4, the typical V/U-shaped LiM curve fades as inputs approach the maximum context size, suggesting that models shift how they utilize input when reaching full capacity. Rather than distinct positional preferences (e.g., primacy or recency), performance increasingly reflects a distance-based bias: relevant information closer to the end yields higher accuracy. In reasoning with Llama-3-70B, Figure 4 shows that while both first and last positions are initially favored, accuracy at the first drops sharply beyond the halfway mark, whereas the last remains stable. At full input length, the model ranks positions as: last > middle > first. This shift illustrates how primacy effects diminish with longer inputs, while later positions retain their advantage. Similar trends hold across other models: all exhibit stronger recency and weaker primacy biases as inputs grow longer. The distance-based bias is especially pronounced in Llama-3-70B, Llama-3.1-70B, and Qwen-2.5-32B. At full length ($L_{\text{rel}} = 1$), performance at middle and last positions even surpasses that at shorter lengths ($0.5 \leq L_{\text{rel}} < 1$) in reasoning. Wu et al. (2025) imply a connection

| Model | $P(\text{RA} = 1 \mid \text{RT} = 0)$ | $P(\text{RA} = 1 \mid \text{RT} = 1)$ | Difference | % Increase |
|---|---|---|---|---|
| Llama-3.1-70B | 0.42 | **0.76** | 0.34 | 81 |
| Llama-3.3-70B | 0.35 | **0.74** | 0.39 | 111 |
| Llama-3-70B | 0.52 | **0.76** | 0.24 | 46 |
| Mistral-Small-24B | 0.39 | **0.72** | 0.33 | 85 |
| Qwen-2.5-32B | 0.38 | **0.82** | 0.44 | 116 |
| Gemma-2-27B | 0.50 | **0.79** | 0.29 | 58 |

Table 3: **Retrieval success substantially boosts reasoning accuracy across models.** We report the reasoning accuracy $P(\text{RA} = 1)$ conditioned on whether retrieval was successful ($\text{RT} = 1$) or not ($\text{RT} = 0$), alongside the absolute difference and percentage increase in accuracy. Values shown are averaged across all datasets.

between LiM and positional biases in training data. Based on that, we hypothesize that the LiM's prominence at shorter input lengths may result from models being pretrained more heavily on short sequences, or from such biases being more common in shorter texts. We show that the observed positional bias trends generalize to real-world tasks by replicating our experiments on a retrieval-augmented QA dataset (Figure 10, Appendix A.2).

### 4.3 Reasoning Failures correlate with Positional Biases in Retrieval

Sections 4.1 and 4.2 showed how LiM, primacy, and recency biases evolve as input lengths approach a model's context window size. We observed that these positional effects appear in both retrieval and reasoning, following strikingly similar patterns. Retrieval and reasoning are closely connected: reasoning over long inputs typically requires first locating the relevant information, a process also described as retrieval. This prerequisite role of retrieval in reasoning, combined with the shared positional patterns shown previously, raises two key questions: (i) How does retrieval success or failure impact reasoning accuracy? and (ii) To what extent are positional biases in reasoning propagated through retrieval?

**Evaluation.** To investigate the relationship between retrieval $RT$ and reasoning $RA$, we compare the conditional probabilities of correct reasoning given failed retrieval, $P(\text{RA} = 1 \mid \text{RT} = 0)$, and correct reasoning given successful retrieval, $P(\text{RA} = 1 \mid \text{RT} = 1)$. If successful retrieval is indeed a required precursor of successful reasoning, we expect a significant and constant performance increase of reasoning in case of sucessful retrieval, i.e. $P(\text{RA} = 1 \mid \text{RT} = 1) > P(\text{RA} = 1 \mid \text{RT} = 0)$. Moreover, $P(RA = 1 \mid RT = 1)$ offers a measure of reasoning performance, disentangled from retrieval errors. If positional biases in reasoning are substantially propagated through retrieval, then disentangling from retrieval errors should reduce or eliminate these biases in reasoning. Conversely, reasoning accuracy conditioned on failed retrieval $P(RA = 1 \mid RT = 0)$ should exhibit stronger positional effects, as it still would inherit position-dependent errors the retrieval stage introduces. We, therefore, use the contrast between these two conditional probabilities to quantify both the impact of retrieval on reasoning accuracy and the extent to which positional biases are propagated through the retrieval stage.

**Results.** As shown in Table 3, correct retrieval boosts reasoning accuracy by up to 116%, a pattern consistent not only on average but also across datasets (Table 4, Appendix A.4). Since all base instances are explicitly controlled to have equal complexity within the same dataset, it is unlikely that the consistent advantage of $P(\text{RA} = 1 \mid \text{RT} = 1)$ stems from corresponding instances being inherently easier. Referring back to question (i) how retrieval success or failure impacts reasoning performance, these results provide strong evidence that retrieval is not merely correlated with, but supports, accurate reasoning in LLMs.

Beyond showing that successful retrieval supports accurate reasoning, we asked (ii) to what extent positional biases in reasoning are propagated through retrieval. Figure 3 revealed strikingly similar positional bias trends in both retrieval and reasoning, with reasoning consistently about 10 percentage points less accurate than retrieval. This pattern is confirmed in Figure 4: as $L_{\text{rel}}$ increases, both retrieval and reasoning reduce their primacy bias, while

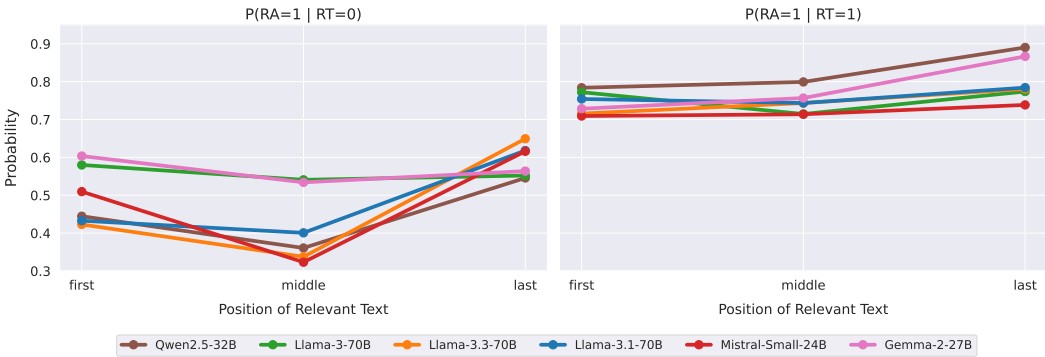

Figure 5: **Positional biases in reasoning conditioned on retrieval accuracy.** Reasoning accuracy $P(RA = 1)$ across positions of relevant text (first, middle, last), conditioned on whether retrieval $RT$ was correct $P(RA = 1 \mid RT = 1)$ (left) or incorrect $P(RA = 1 \mid RT = 1)$ (right). When retrieval fails, models show a pronounced drop in reasoning accuracy for middle-positioned information, a manifestation of the LiM effect. In contrast, when retrieval succeeds, models show higher robustness to positional biases, resulting in almost flat curves for some models (Llama 3.1 70B, Mistral).

recency bias remains stable. These parallels raise the question of whether reasoning's positional biases are inherited from retrieval, with the $\approx 10\%$ performance gap possibly reflecting reasoning's greater intrinsic difficulty. To examine this, we plot $P(\mathrm{RA} = 1 \mid \mathrm{RT} = 0)$ (left) and $P(\mathrm{RA} = 1 \mid \mathrm{RT} = 1)$ (right) in Figure 5 across different positions of relevant text. When retrieval fails, reasoning accuracy exhibits stronger positional biases – most notably a pronounced drop at middle positions (the LiM effect). In contrast, when retrieval succeeds, accuracy remains much more stable across positions, and for some models, such as Mistral-Small-24B and Llama-3.3-70B, the curves are nearly flat, indicating minimal positional biases. These findings suggest that positional biases in reasoning are largely – though not entirely – propagated through retrieval, as reasoning becomes markedly more robust to position when retrieval succeeds.

## 5 Conclusion

We investigated how positional biases behave as input length reaches the context window size of an LLM. Previous literature is mixed on whether these positional biases exist for diverse LLMs. To address this, we conducted an empirical analysis in which input length is defined relative to a model's context window rather than as a fixed variable. In addition, we developed metrics that enable the measurement of positional biases across these relative input lengths. Our analysis shows that considering positional biases with respect to the relative input length is crucial for identifying trends across models with varying context window sizes. Specifically, we found that the Lost in the Middle (LiM) effect, where performance drops when relevant information appears in the middle of the input, consistently occurs for input lengths up to about 50% of a model's context window. Beyond this threshold, the LiM effect diminishes, primarily because the primacy bias tends to fade as input length increases, countering the emergence of the LiM effect. As input length approaches a model's maximum context capacity, we observe an emerging distance-based bias: model performance tends to be higher for information appearing closer to the end of the input sequence. Moreover, our evaluation using retrieval-reasoning minimal pairs, where a retrieval and a reasoning question are posed for the same problem, indicates a strong correlation between failures in retrieval and reasoning. We show that positional biases on reasoning are largely inherited from those on retrieval, demonstrating that retrieval is a precursor of long-context reasoning. Overall, our findings provide valuable insights into how language models process long inputs, up to a point where these inputs actually max out a model's context capacity, and they offer new evaluation protocols for understanding and improving the performance of long-context models.

## Acknowledgments

Funded by the Deutsche Forschungsgemeinschaft (DFG, German Research Foundation) – GRK 2853/1 "Neuroexplicit Models of Language, Vision, and Action" - project number 471607914.

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

# A  Appendix

## A.1  Full-Length Examples of Base Instances

The datasets MonoRel, PIR, and Simplified RuleTaker are taken from Levy et al. (2024), while BoxTracker is generated using the framework proposed by Kim & Schuster (2023). For each dataset, we present a true-labeled example on the left and a false-labeled example on the right. The reasoning questions are taken from the original datasets, while the corresponding retrieval questions were constructed to form reasoning–retrieval minimal pairs. Each retrieval question isolates the factual information required to answer the associated reasoning question. While we showed shortened examples in the main paper (see Table 1), we present the full-length instances below – i.e., premises (in bold) embedded within short narratives.

**Julie Baker is younger than Julian Barton.** This is a fact that remains constant, unchanging like the northern star. It's a truth that is as clear as day, as undeniable as the sky is blue. This is a reality that they both live with, a fact that shapes their interactions and their relationship. It's a detail that is woven into the fabric of their lives, a thread that runs through the tapestry of their shared experiences. It's a simple truth, a straightforward fact, but it's one that bears repeating: Julie Baker is younger than Julian Barton. It's a fact that is as unalterable as the course of the sun across the sky, as immutable as the turning of the earth.
**Samantha Arnold is younger than Julie Baker.** This is a fact that has been established and is well known among their circle of friends and family. It's not a matter of debate or speculation, but a simple truth that Samantha Arnold is younger than Julie Baker. Their age difference is not significant, but it's enough to make Julie Baker feel a sense of responsibility towards Samantha Arnold.

**Dylan Oconnor is slower than Jason Campbell.** This is a fact that has been observed time and time again. Whether it's in their daily routines or in their work, Jason Campbell always seems to be one step ahead. It's not that Dylan Oconnor is particularly slow, it's just that Jason Campbell is exceptionally fast. When they're given a task, Jason Campbell is always the first one to complete it. It's not a matter of skill or intelligence, but rather a difference in pace.
**Jason Campbell is slower than Charles Peters.** This is a fact that has been established time and time again. Whether it's in the morning when they're getting ready for work, or in the evening when they're preparing dinner, Charles Peters always finishes first. He's methodical, meticulous, and yes, slower. It's not that he's lazy or lacks motivation, of course, but if it were, Charles Peters would be the clear winner. She's just faster. Jason Campbell is slower than Charles Peters. It's as simple as that. No matter the task, no matter the day, Charles Peters is always done before Jason Campbell. It's a pattern, a rhythm of their lives.

Reasoning Question: Is Samantha Arnold younger than Julian Barton?
Retrieval Question: Is Samantha Arnold younger than Julie Baker?

Reasoning Question: Is Charles Peters slower than Dylan Oconnor?
Retrieval Question: Is Jason Campbell slower than Dylan Oconnor?

Figure 6: **MonoRel.**

**John's grand ballroom is blue walled**, a truth that resonates with the very foundation of the structure in which it is housed. The moment one sets foot within its confines, it is unmistakably clear that John's grand ballroom is blue walled, with every element of its design and ambiance reinforcing this reality. It is not simply an observation made by the occasional visitor; rather, it is a well-documented and universally accepted fact that John's grand ballroom is blue walled. The consistency with which this fact is presented is remarkable, as every description, every brochure, and every mention of John's grand ballroom invariably highlights that John's grand ballroom is blue walled. It is a detail that has been meticulously planned and executed, ensuring that John's grand ballroom is blue walled, a concept that was integral to its inception and has been faithfully maintained throughout its existence.
**Bridget Burke is in John's grand ballroom**, a statement that has been confirmed repeatedly and has become a well-acknowledged fact within the context it pertains to. The reality that Bridget Burke is in John's grand ballroom is not a temporary condition but a persistent and enduring state of affairs. It is widely recognized that the association between Bridget Burke and John's grand ballroom is so strong that the mention of John's grand ballroom immediately brings to mind the fact that Bridget Burke is there. The constancy of the situation, where Bridget Burke is in John's grand ballroom, is something that has been observed and noted by all who are privy to this information. It is a truth that has been spoken of so often that it has taken on the weight of a mantra: Bridget Burke is in John's grand ballroom.

**John's living room is blue walled**, a statement that reverberates with unwavering certainty throughout the entire establishment. The moment one steps into the vicinity, it is palpable that John's living room is blue walled, with every element within its confines meticulously aligned to affirm blue walled. It is not a mere observation that John's living room is blue walled; it is a fact that is deeply ingrained in the consciousness of everyone associated with the place. The architects and designers have left no stone unturned in ensuring that John's living room is blue walled, a sentiment that is echoed in every corner of the room. This is not an accidental feature; it is a deliberate design choice that has been emphasized repeatedly, making it clear that John's living room is blue walled.
**Matthew Ortiz is in John's living room**, a fact that has become as much a part of the place as the walls and the ceiling. The truth that Matthew Ortiz is in John's living room is so well-established that it is almost redundant to mention it, yet it is mentioned, again and again, a testament to its unassailable veracity. There is a certain rhythm to the repetition, a cadence that reinforces the knowledge that Matthew Ortiz is in John's living room with every iteration. It is a reality that has settled into the consciousness of all who are aware of the space, a steady drumbeat that resonates with the phrase: Matthew Ortiz is in John's living room

Reasoning Question: Is Bridget Burke in a blue walled room?
Retrieval Question: Is Bridget Burke in John's grand ballroom?

Reasoning Question: Is Matthew Ortiz in a red walled room?
Retrieval Question: Is John's living room red walled?

Figure 7: **PIR.**

Julia found two Boxes: Box 0 and Box 1. Box 0 contains a cross. Box 1 contains a gift. Julia puts the cake into Box 0. Julia moves the contents of Box 0 to Box 1.

Silvie found two Boxes: Box 0 and Box 1. Box 0 contains a gift. Box 1 contains a cross. Silvie moves the contents of Box 0 to Box 1. Silvie puts the coat into Box 0.

Reasoning Question: After performing all operations, Box 1 contains the cross and the cake and the gift.
Retrieval Question: The cake was put into Box 0.

Reasoning Question: After performing all operations, Box 1 contains the coat.
Retrieval Question: The gift was put into Box 1.

Figure 9: **BoxTracker.**

| If X is good and X is small then X is loud. | If X is tall and X is big then X is kind. |
|---|---|
| **Dave is small**. Dave, being small, often finds himself having to navigate through a world that is not always accommodating to his size. Whether it's struggling to reach items on high shelves or being overlooked in a crowd, Dave's small stature is a constant presence in his daily life. Despite the challenges he faces, Dave has learned to adapt and make the most of his petite frame. He has become an expert at finding creative solutions to everyday problems, using step stools and reaching tools to access things that are out of his reach. Dave's smallness has also given him a unique perspective on the world around him.

**Dave is good.** Dave is known for his goodness. He has always been good, and everyone around him can attest to that. From a young age, Dave displayed a natural inclination towards kindness and compassion. His friends and family have always admired his ability to see the good in others and to lend a helping hand whenever needed. Dave's goodness is not just limited to his personal relationships; it extends to his professional life as well. In the workplace, Dave is known for his integrity and his willingness to go above and beyond to ensure the success of his team. His colleagues often seek his advice and guidance, knowing that he will always provide a fair and thoughtful perspective | **Dave is tall**. Dave, who is tall, is known for his towering height. Standing at an impressive height, Dave's tall stature is hard to miss. People often comment on how tall Dave is, as it is a defining characteristic of his appearance. With his tall frame, Dave stands head and shoulders above the crowd, quite literally. Whether he is walking down the street or attending social gatherings, Dave's tallness is always noticeable. His tallness has become a part of his identity, and it is something that people have come to expect from him. Dave's tallness is not just a one-time observation; it is a consistent trait that he possesses.

**Dave is red**. Dave, the person we all know and love, is red. Yes, you heard it right, Dave is red. It's not something out of the ordinary, it's just a simple fact that Dave is red. From his head to his toes, Dave is red. It's a characteristic that defines him, that sets him apart from the rest. When you see Dave walking down the street, you can't help but notice his redness. It's become a part of his identity, a part of who he is. People have come to expect it, to accept it as the usual. Dave's redness is not something that surprises anyone anymore |
| Reasoning Question: Dave is loud.
Retrieval Question: Dave is small and good. | Reasoning Question: Dave is kind.
Retrieval Question: Dave is tall and big. |

Figure 8: **Simplified RuleTaker.**

## A.2 Additional Experiments

### A.2.1 Real-World Task

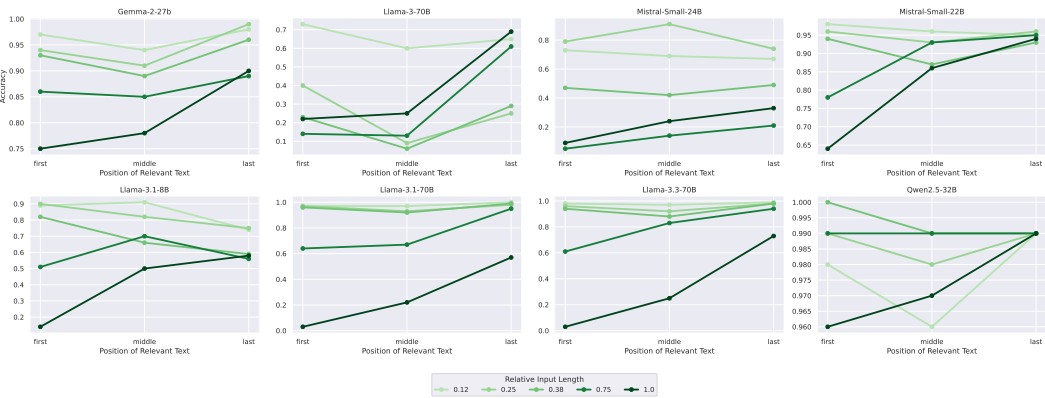

Figure 10: **Retrieval-Augmented QA.** To assess whether the trends observed in this paper extend to real-world settings, we constructed a retrieval-based QA dataset. We selected long-tail factual questions (Veseli et al., 2023) that a set of models (Gemma-2-27B, Llama-3-70B, Mistral) fail to answer without supporting evidence but succeed with access to the corresponding Wikipedia page. This preselection ensures that the answer depends on retrieval rather than memorized knowledge from pre-training, mirroring the controlled conditions of our synthetic datasets. Each instance consists of the answer-containing Wikipedia page concatenated with irrelevant ones, with only one document per input providing the necessary information (similar to (Liu et al., 2024)). Unlike our binary tasks, the model here must generate the object of a (subject, relation, object) triple (e.g., (France, hasCapital, Paris)) where the object typically spans multiple tokens. As models varied in how consistently they followed the instruction to output only the object, we report whether the ground truth object appears anywhere in the answer instead of the exact match accuracy. We follow the same setup as in the main paper: varying the position of the relevant document (first, middle, last) and input length (5 levels), resulting in 1500 instances (100 questions × 3 positions × 5 lengths). We included Mistral-Small-22B and Llama-3.1-8B to compare them with ntheir ewer and larger versions (Mistral-Small-24B, Llama-3.1-70B). Performance trends align with the synthetic setting: accuracy declines when relevant information appears earlier in longer inputs, and the LiM effect is most prominent in shorter contexts.

### A.2.2 Comparing Relative Input Lengths

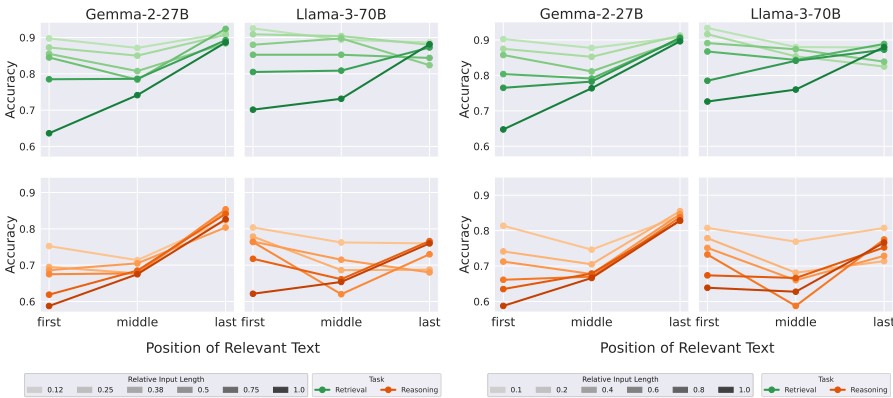

Figure 11: **Comparison between different $L_{rel}$.** We compare the different relative input lengths $L_{\text{rel}} = \{0.12, 0.25, 0.38, 0.5, 0.75, 1.0\}$ with $L_{\text{rel}} = \{0.1, 0.2, 0.4, 0.6, 0.8, 1.0\}$, and observe no difference in the overall trends.

### A.2.3 Comparing Model Sizes

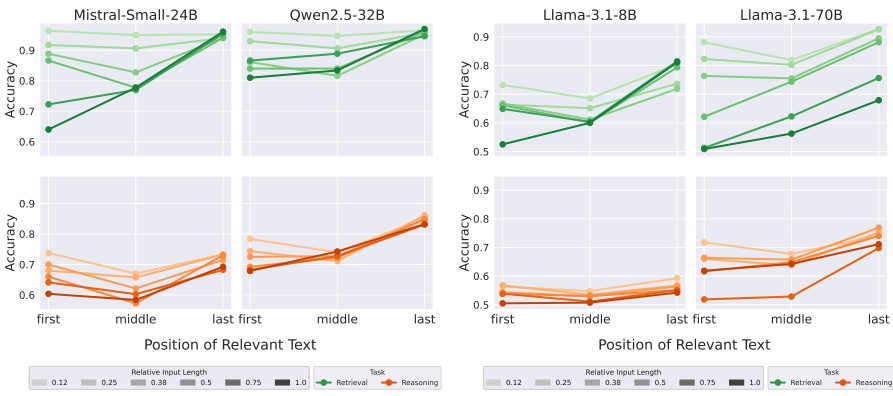

Figure 12: **Comparison across models of different sizes with matched context lengths.** On the left, we show Mistral-Small-24B and Qwen2.5-32B (both with 32K-token context windows); on the right, Llama-3.1-8B and Llama-3.1-70B (both with 128K-token windows). Despite the differences in model scale, we observe similar positional trends within each context length setting: performance drops when relevant information appears earlier in the input as input length increases and the LiM is most noticeable for shorter input lengths.

### A.3 Dataset-specific Visualizations

In this section we show visualizations on model and dataset difference without averaging. Overall, our findings on positional bias are consistent across both models and datasets, even when examining individual conditions. Retrieval tasks show highly stable patterns – i.e., a Lost in the Middle (LiM) effect for relative input lengths up to 0.5 and steep primacy drops beyond that point. This likely reflects the relatively uniform complexity of retrieval tasks, which does not vary substantially across datasets. Reasoning tasks also display consistent positional trends, but detecting these effects becomes more difficult when model performance approaches chance level, as random guessing can obscure sensitivity to input position. This issue is most evident in more complex datasets like Simplified RuleTaker and Box.

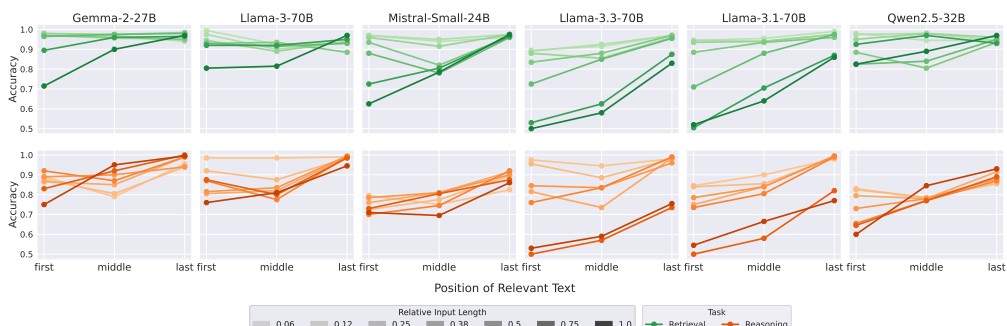

Figure 13: **MonoRel.**

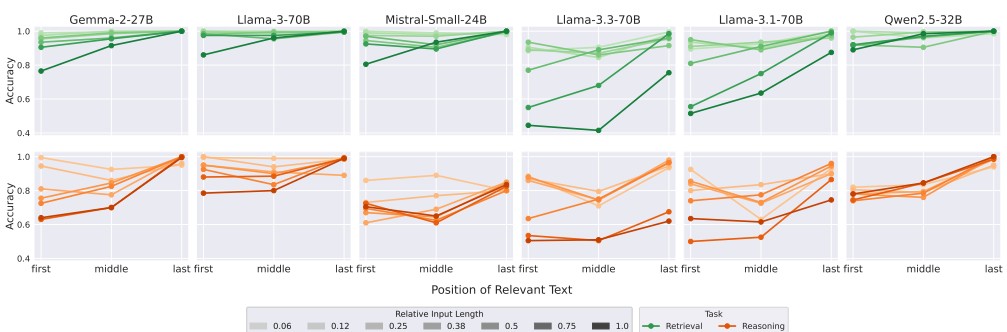

Figure 14: **PIR.**

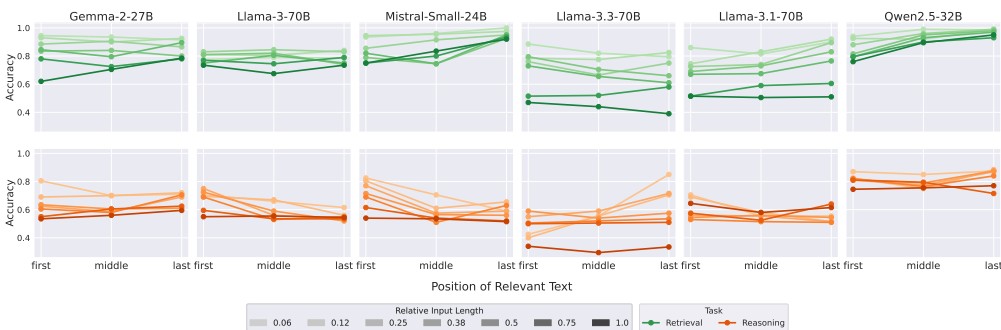

Figure 15: **Simplified RuleTaker.**

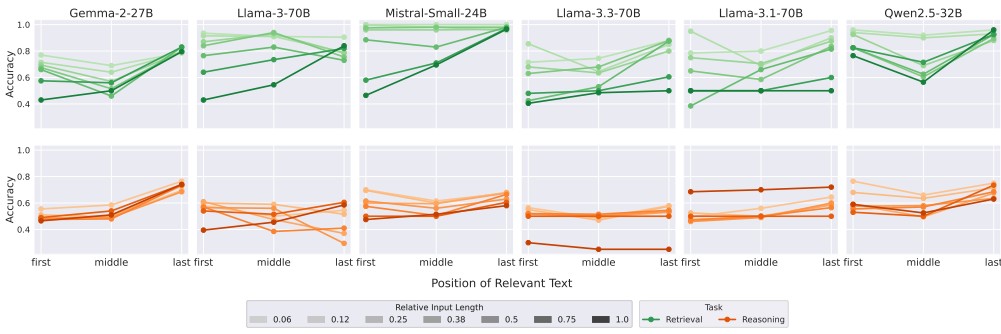

Figure 16: **BoxTracker.**

## A.4 Conditional Probabilties

| Model | Dataset | P(RA=1|RT=0) | P(RA=1|RT=1) | Difference | ~Increase (%) |
|---|---|---|---|---|---|
| Llama-3.1-70B | MonoRel | 0.56 | **0.85** | 0.29 | 52 |
| | PIR | 0.57 | **0.81** | 0.24 | 42 |
| | RuleTaker | 0.25 | **0.70** | 0.45 | 180 |
| | BoxTracker | 0.30 | **0.66** | 0.36 | 120 |
| Llama-3.3-70B | MonoRel | 0.49 | **0.88** | 0.39 | 80 |
| | PIR | 0.39 | **0.84** | 0.45 | 115 |
| | RuleTaker | 0.15 | **0.71** | 0.56 | 373 |
| | BoxTracker | 0.36 | **0.55** | 0.19 | 53 |
| Llama-3-70B | MonoRel | 0.73 | **0.91** | 0.18 | 25 |
| | PIR | 0.49 | **0.94** | 0.45 | 92 |
| | RuleTaker | 0.45 | **0.64** | 0.19 | 42 |
| | BoxTracker | 0.42 | **0.53** | 0.11 | 26 |
| Gemma-2-27B | MonoRel | 0.62 | **0.92** | 0.30 | 49 |
| | PIR | 0.60 | **0.87** | 0.27 | 45 |
| | RuleTaker | 0.47 | **0.67** | 0.20 | 43 |
| | BoxTracker | 0.34 | **0.70** | 0.36 | 106 |
| Mistral-Small-24B | MonoRel | 0.49 | **0.83** | 0.34 | 0.69 |
| | PIR | 0.27 | **0.76** | 0.49 | 181 |
| | RuleTaker | 0.42 | **0.65** | 0.23 | 55 |
| | BoxTracker | 0.39 | **0.62** | 0.23 | 59 |
| Qwen2.5-32B | MonoRel | 0.29 | **0.84** | 0.55 | 189 |
| | PIR | 0.34 | **0.87** | 0.53 | 156 |
| | RuleTaker | 0.61 | **0.83** | 0.22 | 36 |
| | BoxTracker | 0.29 | **0.76** | 0.47 | 162 |

Table 4: Reasoning accuracy (RA = 1) conditioned on retrieval success (RT = 1) versus failure (RT = 0) across datasets and models. In balanced binary tasks (as is the case here), P(RA=1 | RT=0) values near 0.5 indicate chance-level performance. Across all models and datasets, reasoning accuracy increases substantially when retrieval succeeds - rising by at least 25% and up to 373% - highlighting the critical role of retrieval in enabling correct reasoning. P(RA = 1 | RT = 1) consistently being higher might but is unlikely to stem from the corresponding instances being inherently easier. The base instances are artificially generated and therefore control for complexity: all examples within the same dataset follow the same logical structure and reasoning depth, with identical distributions of operations and entities. This supports the interpretation that successful retrieval itself is the key driver of improved reasoning performance.

