# OpenReview forum: "Positional Biases Shift as Inputs Approach Context Window Limits"
_colmweb.org/COLM/2025/Conference — COLM 2025_

### Official Review · Reviewer_DcAC · 2025-04-19

**Rating:** 6
**Confidence:** 4
**Ethics Flag:** 1

**Summary:**

This paper investigates positional biases in LLMs, focusing on the "Lost in the Middle" effect, where models struggle with middle-positioned information. By defining input length relative to a model’s context window, the authors show LiM dominates when inputs occupy ≤50% of the context window, but diminishes beyond this threshold as primacy bias weakens and a distance-based bias emerges. They also demonstrate a strong correlation between retrieval failures and reasoning errors. These findings reconcile inconsistent prior results and highlight the importance of evaluating positional effects relative to model capacity.

**Questions To Authors:**

* I am wondering why the relevant information’s position is not evenly spaced in investigation (e.g. 0.2 0.4 0.6, etc.) but in current way shown in Figure 2’s caption and Section 3.2 Line 155
*  Is current evaluation data representative of long-context reasoning and retrieval tasks? For instance, there may exist even more complex tasks that may involve multiple hops of reasoning (which of course is more difficult)
* Current model used is mainly based on context length. Will the model family / model scale itself show different trends in results?
* The 50% threshold mentioned in paper seems is from empirical results, but is there any explanation why the threshold is 50%? How to attribute this phenomenon and what does it imply?
* LiM here is seen as a relative concept. As from equation 1, the LiMi score is relative to acc^F and acc^L. However, from figure 1, the middle position performance indeed becomes lower and lower as relative input length increases. Could this be really call “Lost in the middle” diminishes? As the performance does not rise up — it’s only the decrease in beginning performance that changes the shape of the curve.

**Reasons To Accept:**

* The paper introduces a novel relative input length framework to systematically analyze positional biases across models with varying context windows.
* The paper provides comprehensive validation using multiple models and datasets, resolving contradictions in prior research.

**Reasons To Reject:**

* The paper is limited to synthetic or modified datasets, raising concerns about generalizability to real-world long-context tasks.
* The paper also does not establishes causality between retrieval failures and reasoning errors.
* The paper lacks deeper analysis on the cause and implication of the phenomenon discovered.

---

> ### Author Response · Authors · 2025-06-03
>
> Thank you for your thoughtful comments.
>
> **Generalizability to real-world tasks and new results.** The use of synthetic datasets is a precedent that was set by previous work Levy et al. 2024; we follow this precedent in our paper. The advantage of synthetic data is that (a) we can precisely control the position of the relevant information and (b) the model cannot rely on knowledge memorized in pretraining. **New results:** Nonetheless, to investigate the questioned generalizability, we constructed a dataset by concatenating Wikipedia pages, such that only one of the pages could answer each retrieval question. Addressing point b) above, we precede this with a test if the model is able to answer the question without the wikipedia page via its in-weight knowledge and discard such cases. This yielded a test set of 1500 samples. The findings are consistent with those on the synthetic datasets in the paper (see new material [Section 3](https://tinyurl.com/4742d56f)), suggesting that the same mechanisms are at play in more real-world settings.
>
> **Causality of retrieval and reasoning errors.** Our main goal was to uncover how relative input length drives positional biases. During that investigation we noticed a close relation of retrieval- and reasoning positional biases (lines 302-306 in the manuscript). This empirical overlap suggested that retrieval failures might propagate directly to reasoning mistakes (e.g. Karpinska et al., 2024; Levy et al., 2024), motivating an additional strictly supplementary causality check for context.
>
> Following the reviewer’s suggestion, we measured P(Reasoning=correct | Retrieval=incorrect), i.e., the probability of a correct reasoning answer when the corresponding retrieval answer is wrong. If retrieval truly gates reasoning, this conditional accuracy should sit near the 0.5 chance level of our balanced binary tasks. The conditional probabilities for Llama 3 70B, Gemma 2 27B, Mistrall Small 3 24B and Llama 3.1 8B are 0.54, 0.48, 0.44 and 0.38, respectively. While Llama 3.1 8B shows a lower conditional accuracy, the values lie around the 0.5 baseline, supporting the hypothesis that retrieval failures largely determine reasoning errors. While this causality check is secondary to our main focus on input-length effects, it reinforces the interpretation that the positional biases observed in reasoning are downstream of those in retrieval.
>
> **Deeper analysis of the cause of the phenomenon and its implication.** Our primary goal was to explain discrepancies in prior work on positional biases effect through systematic empirical analysis. Understanding why such patterns occur first requires robust testing - but evaluating models at full context length is resource-intensive (e.g., ~$10,000 for GPT-4 Turbo; 80 GPU-hours on 4xH100 per dataset for Llama 3.1–70B). These resource and time demands naturally limit the scope of any single study. Our findings have concrete implications for model evaluation and deployment: by introducing relative input length, we clarify when positional biases emerge, enabling more reliable benchmarking and better model choices for long-context tasks like retrieval-augmented QA.
>
> That said, we agree with the reviewer that understanding how relative input length affects LiM is a valuable next step and will add a discussion in the paper. A plausible explanation, supported by [Wu et al. (2025)](https://arxiv.org/pdf/2502.01951), is that the LiM effect specifically arises from training data where information often appears at the beginning or end. Since models like Llama 3  (see [Llama Team](https://arxiv.org/pdf/2407.21783)), are first trained on shorter sequences and only gradually adapted to longer ones, the LiM effect in the early context window may be a training artifact. The early training emphasis on shorter inputs could lead to stronger position sensitivity in the first half of the context window - possibly explaining why LiM weakens as input length increases.
>
> **Why not use evenly spaced relative input lengths?** We chose relative input lengths (0.06, 0.12, 0.25, 0.38,...)  to align with Levy et al. (2024), where absolute input lengths (500–3000 tokens) map to our relative spacings (0.06–0.38) for an 8K context window, enabling direct comparison. We also tested evenly spaced values (e.g., 0.1, 0.2, 0.4, 0.6, 0.8, 1) and found the qualitative patterns unchanged (see new material [Section 3.1](https://tinyurl.com/ru3wj84a)); we will include these results in the paper.
>
> **Comparison between model families and sizes.** Please refer to the general response.
>
> **LiM as a relative concept.** We follow the original definition of LiM from Liu et al. 2023 by describing it as the difference between accuracy in the middle and at the beginning/end. LiM does not disappear because the middle gets better, but rather disappears even though middle accuracy also declines - driven by a collapsing primacy bias. See the corresponding remark in lines 269 ff of the main paper.

---

> > ### Comment · Reviewer_DcAC · 2025-06-04
> >
> > Thanks the author for the response and additional experiments. Most of my concerns are solved. Hope to see the incorporation of new results and some of your hypothesized insights in the next version of your paper. I will raise the overall score to 6.

---

### Official Review · Reviewer_db9o · 2025-05-11

**Rating:** 6
**Confidence:** 3
**Ethics Flag:** 1

**Summary:**

The paper conducted a comprehensive analysis that uses an input length relative to a model’s context window size, rather than the length itself as a fixed variable. The findings are that the relative input length plays a crucial role in positional biases. Specifically, it was observed that the LiM effect is prominent when input sequences occupy up to 50% of a model’s context window. Beyond that, a primacy bias is gradually overshadowed by a distance-based bias, effectively eliminating the LiM effect.

**Questions To Authors:**

We sometimes found strange citation formats in the draft. Please try to properly use \cite rather than \citet, when necessary.

**Reasons To Accept:**

The paper conducted a comprehensive analysis that uses an input length relative to a model’s context window size, rather than the length itself as a fixed variable. The findings in the paper can reconcile the inconsistency in the findings in the previous work. While it is natural and simple, utilizing an input length relative to a model’s context window size is reasonable and can yield interesting results.

**Reasons To Reject:**

While the authors showed only averaged results across datasets and models in Sec. 4.1, from Fig. 5 in Sec. 4.2, different models look having different characteristics in the results. So, discussing the differences among the models would be valuable and fruitful. Furthermore, the results across different datasets might have different characteristics. So, I hope the authors will investigate and discuss the differences, if any. They can be shown in the appendices.

---

> ### Author Response · Authors · 2025-06-03
>
> **Elaborate on model and dataset differences**. Thank you for the suggestion - we will, of course, include all detailed (non-averaged) results (see new material [Section 2](https://docs.google.com/document/d/14LlX7CDpMRyh_qfPEaXVBvpriatdwUaIwttCruOXQQ8/edit?tab=t.0#bookmark=id.h8hjlmbg9j6t), p. 2-4) in the appendix and add a discussion of significant differences in the manuscript.
> Overall, our findings on positional bias are consistent across both models and datasets, even when examining individual conditions. Retrieval tasks show highly stable patterns—i.e., a Lost in the Middle (LiM) effect for relative input lengths up to 0.5 and steep primacy drops beyond that point. This likely reflects the relatively uniform complexity of retrieval tasks, which does not vary substantially across datasets. Reasoning tasks also display consistent positional trends, but detecting these effects becomes more difficult when model performance approaches chance level, as random guessing can hide sensitivity to input position. This issue is most evident in more complex datasets like Simplified RuleTaker and Box. Nevertheless, we still observe strong recency biases and sharp primacy drops with increasing input length, confirming the robustness of our findings. We will add this discussion to our manuscript.

---

> > ### Comment · Reviewer_db9o · 2025-06-05
> >
> > Thank you for the responses. I will keep my score.

---

### Official Review · Reviewer_rJZo · 2025-05-13

**Rating:** 6
**Confidence:** 4
**Ethics Flag:** 1

**Summary:**

This work investigates the positional biases of LLMs in different relative input lengths.
Four LLMs with different context window sizes are involved in experiments, including Gemma 2 27B (8K), Llama 3-70B (8K), Mistral-Small-3-24B (32K), and Llama 3.1 8B (128K).
A metric, LiMi, is introduced to measure the intensity of the ``Lost in Middle'' (LiM) phenomena of positional biased.
Two types of tasks (retrieval and reasoning) on four datasets are evaluated in experiments.
Experimental results confirm the Lost in Middle effect, and also point out the raise of distance bias as the increase of input length.

**Questions To Authors:**

1. Wrong symbol "¿" in the caption of Table 1.

**Reasons To Accept:**

1. This work analyzes the problem of positional biases in different input lengths, providing another perspective to this serious issue that is still yet to tackle in current LLMs.
2. A quantitative approach is proposed to evaluate the Lost in Middle effect. Interesting findings are also shown and discussed.

**Reasons To Reject:**

1. The choice of the models in experiments could be more thoughtful. In the current version, the four models are chosen from different sizes (number of parameters), generations, manufacturers (Google, Meta, Mistral), and context window lengths. It could be more analytical to compare the models with some dimensions fixed. For example, it is interesting to know the comparison between Llama 3.1 8B and Llama 3.1 70B as both are the same generation, same claim context window size but different in model sizes and capabilities. It is also interesting to compare Llama 3.1 70B with Llama 3.3 70B. In other words, the chosen and the comparison of the models could be organized in a more systematic and thoughtful manner.

---

> ### Author Response · Authors · 2025-06-03
>
> **More systematic selection of models.** We appreciate your feedback. Please see *evaluation on more models* in the general response and for visualizations of the requested experiments see new material [here](https://docs.google.com/document/d/14LlX7CDpMRyh_qfPEaXVBvpriatdwUaIwttCruOXQQ8/edit?tab=t.0#bookmark=id.fubv43o6td19).

---

> > ### Comment · Reviewer_rJZo · 2025-06-07
> >
> > Thank you for the update. I raise my score accordingly.

---

### Author Response · Authors · 2025-06-03

We thank the reviewers for their thoughtful comments and will take them very seriously in revising our paper.

First we want to address a point that a few reviewers made: the **evaluation on more models**. We specifically chose models of different context lengths to establish our main finding: that the driving factor in the strength of positional biases is input size relative to context length. In doing this, we also varied the model size and model family to ensure that our findings are robust across LLMs.

While it is not the main point we wanted to make in this paper, we agree with reviewers R1 (rJZo ) and R3 (DcAC) that it would also be interesting to investigate the strength of positional biases as a function of model size, family or generation. We have extended our experiments (see new material [Section 1](https://docs.google.com/document/d/14LlX7CDpMRyh_qfPEaXVBvpriatdwUaIwttCruOXQQ8/edit?tab=t.0#bookmark=id.fubv43o6td19)) to also include
- Generation Comparison: Llama **3.1** 70B (128K) and Llama **3.3** 70B (128K)
- Size Comparison: Llama 3.1 **70B** and Llama 3.1 **8B**
- Model Family Comparison: **Mistral** 24B  and **Qwen**2.5 32B

We find that model size, family and generation do not significantly affect positional bias trends - though, as expected, larger models perform better overall. More specifically, our key finding - that positional biases depend primarily on relative context window size - is validated and consistent across the study.

---

### Decision · Program_Chairs · 2025-07-08

**Decision:**

Accept

**Comment:**

This paper presents a thorough empirical study of needle-in-the-haystack across different context lengths. Namely, accuracy is studied as a function of both needle position as well as the ratio of the input to model's max context length. The findings are interesting, that lost-in-the-middle is consistent but also gives way to lost-everywhere-except-the-end as inputs reach the max context length. This intuitively makes sense, as data used for long-context extension likely has a strong recency bias as well.

The reviews are consistently positive-leaning. I see no reason to reject this paper, as the measurement and evaluation presented is the most important bit. Fixing the issue this paper finds is likely a question of data.

Comments
1. Typo in Section 4.1 Title: "Dispite"
2. Appendix A is empty
3. Figure 5 Caption: Should this be disappearing primacy bias?